# PreciseCache: Precise Feature Caching for Efficient and High-fidelity Video Generation

**Jiangshan Wang**[1,2,3]    **Kang Zhao**[3]    **Jiayi Guo**[2]    **Jiayu Wang**[3]
**Hang Guo**[2]    **Chenyang Zhu**[2]    **Xiu Li**[2†]    **Xiangyu Yue**[1†]
[1]MMLab, CUHK    [2]Tsinghua University    [3]Tongyi Lab, Alibaba

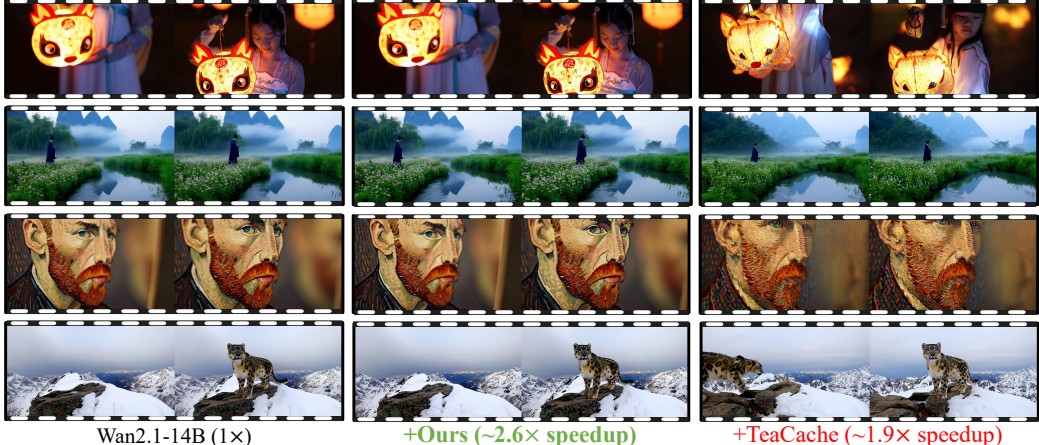

Wan2.1-14B (1×)        +Ours (~2.6× speedup)        +TeaCache (~1.9× speedup)

Figure 1: **Qualitative Results of PreciseCache on Wan2.1-14B** (Wang et al., 2025). Compared with previous methods, our PreciseCache achieves higher acceleration (about 2.6× speedup) of the base model without sacrificing the quality of generated videos.

## Abstract

High computational costs and slow inference hinder the practical application of video generation models. While prior works accelerate the generation process through feature caching, they often suffer from notable quality degradation. In this work, we reveal that this issue arises from their inability to distinguish truly redundant features, which leads to the unintended skipping of computations on important features. To address this, we propose **PreciseCache**, a plug-and-play framework that precisely detects and skips truly redundant computations, thereby accelerating inference without sacrificing quality. Specifically, PreciseCache contains two components: LFCache for step-wise caching and BlockCache for block-wise caching. For LFCache, we compute the Low-Frequency Difference (LFD) between the prediction features of the current step and those from the previous cached step. Empirically, we observe that LFD serves as an effective measure of step-wise redundancy, accurately detecting highly redundant steps whose computation can be skipped through reusing cached features. To further accelerate generation within each non-skipped step, we propose BlockCache, which precisely detects and skips redundant computations at the block level within the network. Extensive experiments on various backbones demonstrate the effectiveness of our PreciseCache, such as achieving an average of 2.6× speedup on Wan2.1-14B without noticeable quality loss.

## 1 Introduction

Video generation models (Zheng et al., 2024; Yang et al., 2024; Kong et al., 2024; Wang et al., 2025) have demonstrated impressive capabilities in producing high-fidelity and temporally coherent videos. However, they always suffer from extremely slow inference speed, posing a significant challenge to their application. Although some works attempt to alleviate the problem through distillation

---

[†] Corresponding authors.

(Geng et al., 2025; Song et al., 2023), they always need additional training, which is computationally intensive. To address this, feature caching (Zhao et al., 2024b; Liu et al., 2025b; Lv et al., 2024) has emerged as a popular approach to accelerate the process of video generation, which skips the network inference in several denoising steps by reusing the cached features from previous steps. However, these works usually adopt a uniform caching scheme (i.e., performing a full inference every $n$ steps, caching the features, and reusing them until the next full inference), which overlooks the varying importance of different timesteps in determining the output quality, resulting in insufficient speedup or noticeable quality degradation. As a result, some recent works (Liu et al., 2025a; Kahatapitiya et al., 2024; Chu et al., 2025) propose adaptive caching mechanisms that design metrics to adaptively decide whether to perform the full model inference or reuse cached features at each denoising timestep. However, these methods require complicated additional fitting or extensive hyperparameter tuning, and their cache decision criteria remain suboptimal, leading to unsatisfying generated results. Consequently, designing adaptive run-time caching mechanisms that maximizes acceleration while preserving video quality remains challenging.

In this work, we propose **PreciseCache**, an adaptive video generation acceleration framework that precisely identifies redundant features and skips their computation through feature caching, thereby enabling maximal speedup without compromising video quality. To this end, at each denoising step, we analyze the influence of reusing cached features on the final generation quality. The results (Figure 3a) indicate that as the denoising process progresses from high to low noise stages, the influence of reusing cached features gradually diminishes (Figure 2). This is consistent with the intuition that the diffusion process models low-frequency structural information at high-noise steps while refining the generated content with high-frequency details at low-noise steps (Wan, 2025). The structural information is crucial for video generation, while high-frequency details are usually perceptually insignificant, where the computation can be skipped to achieve acceleration. Consequently, we propose **Low-Frequency Difference (LFD)**, which measures

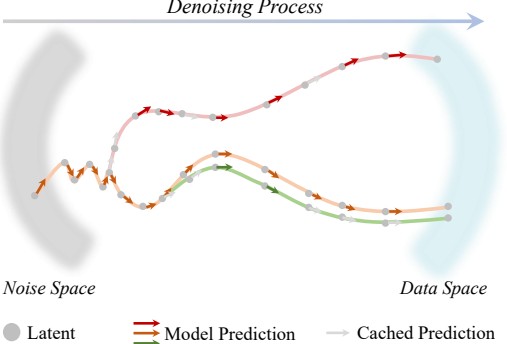

Figure 2: **The Illustration of the Denoising Process for Video Generation**. At high-noise timesteps, the prediction of the model varies significantly. Reusing the cached features in this stage (the red line) can significantly affect both the content and the quality of generated videos compared to the videos generated without caching (the orange line). In contrast, the feature caching during low-noise timesteps only introduces negligible impacts (the green line).

the difference between the low-frequency components of the model's outputs at adjacent denoising steps. Experiments in Figure 4b illustrate that LFD effectively estimates the redundancy of each denoising step (as illustrated in Figure 3a) and therefore can be leveraged to indicate caching.

Based on the above analysis, we propose **LFCache** for step-wise caching. Specifically, at each denoising step, we aim to leverage the LFD between the network prediction at the current step and that at the previous cached step as the criterion to indicate caching. However, directly calculating LFD cannot achieve acceleration because it requires calculating the current-step predictions through the full model inference. To address this challenge, we observe that the LFD exhibits low sensitivity to the resolution of the input latent (Figure 6), suggesting that a lightweight downsampled latent is sufficient for its estimation. Consequently, at each denoising step in our LFCache framework, the noisy latent is firstly downsampled and fed into the model for a quick "trial" inference, obtaining an estimated prediction. The LFD is calculated between this prediction and the cached prediction, which is then used to indicate caching. Due to the reduced latent size, the additional overhead for the inference of the downsampled latent is negligible compared to the overall generation time.

The LFCache identifies and eliminates the redundancy at the timestep level, focusing on the final output of the entire network at each denoising step. Beyond this, we introduce **BlockCache**, which delves into the network and further accelerates the generation process by performing the block-wise caching inside each non-skipped timestep identified by LFCache. Specifically, we assess the redundancy of each transformer block by measuring the difference between its input and output features. Our analysis (Figure 7) reveals that only a subset of transformer blocks make substantial

modifications to the input feature (which we refer to as pivotal blocks), while others have minimal impact (which we refer to as non-pivotal blocks). BlockCache caches and reuses the outputs of these non-pivotal blocks to reduce redundant computation.

We evaluate our PreciseCache on various state-of-the-art video diffusion models, including Opensora (Zheng et al., 2024), HunyuanVideo (Kong et al., 2024), CogVideoX (Yang et al., 2024), and Wan2.1 (Wang et al., 2025). Experimental results demonstrate that our approach can achieve an average of 2.6× speedup while preserving video generation quality, outperforming a wide range of previous caching-based acceleration methods.

## 2 RELATED WORK

### 2.1 VIDEO DIFFUSION MODEL

Diffusion models (Ho et al., 2020; Rombach et al., 2022; Peebles & Xie, 2023) have become the leading paradigm for high-quality generative modeling in recent years. Within the video generation domain, diffusion-based approaches have attracted increasing attention, driven by the rising demand for temporally coherent and high-resolution dynamic content (Blattmann et al., 2023b;a; Hong et al., 2023; Wang et al., 2024b;c). Recent advances have consequently seen a shift from conventional U-Net architectures (Ronneberger et al., 2015) towards more scalable Diffusion Transformers (DiTs) (Peebles & Xie, 2023), which offer enhanced capacity to model intricate temporal dynamics across frames. State-of-the-art DiT-based video diffusion models such as Sora (Brooks et al., 2024; Zheng et al., 2024), CogvideoX (Yang et al., 2024), HunyuanVideo (Kong et al., 2024), and Wan2.1 (Wang et al., 2025) have demonstrated impressive performance in synthesizing coherent and high-fidelity videos. Despite these advancements, the inherently iterative denoising process in diffusion models introduces considerable inference latency, which remains a critical challenge for real-time or large-scale deployment.

### 2.2 DIFFUSION MODEL INFERENCE-TIME ACCELERATION

Distillation or pruning are common methods for model acceleration Gou et al. (2021); Cheng et al. (2024); Liu et al. (2018); Wang et al. (2024d); Fang et al., which is also widely applied in diffusion models(Song et al., 2023; Meng et al., 2023; Sauer et al., 2024; Wang et al., 2026). However, they usually require large-scale training, which is time-consuming and resource-intensive. As an alternative, training-free inference acceleration methods (Song et al., 2021; Karras et al., 2022; Lu et al., 2022; Bolya & Hoffman, 2023; Wang et al., 2024a; Zou et al., 2025; Zhang et al., 2025b;a; Xi et al., 2025; Ye et al., 2024) have gained considerable attention for speeding up diffusion model inference without costly retraining. Among these methods, Feature caching is one of the most popular methods for training-free video generation acceleration, which leverages redundancy across iterative denoising steps. Early static caching methods (Selvaraju et al., 2024; Chen et al., 2024; Zhao et al., 2024b) rely on fixed schemes, but lack flexibility to adapt to varying process dynamics. To overcome this, adaptive caching approaches (Wimbauer et al., 2024; Liu et al., 2025a; Chu et al., 2025; Kahatapitiya et al., 2024; Zhou et al., 2025; Guo et al., 2025; 2026) propose to adaptively decide when to apply the caching and reusing mechanism during the denoising process. However, these methods usually suffer from notable quality degradation or extensive hyperparameter tuning.

## 3 METHOD

In this section, we introduce the **PreciseCache** method in detail. First, we analyze the influence of reusing the cached feature at each timestep on the final generated result, proposing the **Low-Frequency Difference (LFD)** metric to precisely estimate this influence at each timestep during the video generation process. Then, we introduce **LFCache** for timestep-level caching. At each denoising step, our LFCache framework first feeds a downsampled latent into the model, obtaining an estimated output at this step. LFD is calculated between this output and the cached output, which is used to determine whether to apply caching. Finally, we further propose **BlockCache**, which performs the caching and reusing mechanism at the block level within the non-skipped timesteps. The overall algorithm of our PreciseCache is shown in Figure 5 and Algorithm 1.

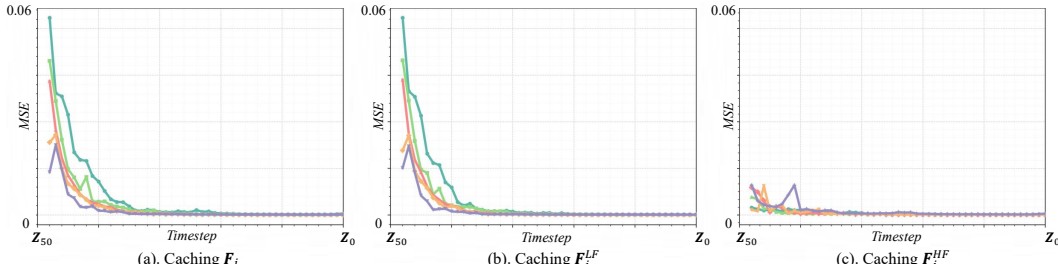

Figure 3: **The Impact of Reusing the Cached Model Prediction at Each Timestep.** Considering a 50-step denoising process from the Gaussian Noise $\boldsymbol{Z}_{50}$ to a clean latent $\boldsymbol{Z}_0$, we respectively reuse the model output at timestep $t_{i+1}$ for each $i \in \{49, 48, \cdots, 0\}$, and perform the subsequent denoising steps to generate the final video. We then compare each resulting video with the baseline (i.e., generated without caching and reusing) to evaluate the impact of reusing cached predictions. Different colors indicate different prompts.

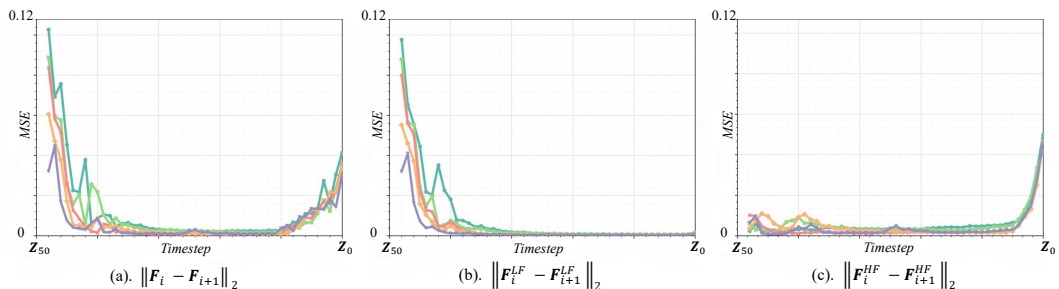

Figure 4: **The Difference between Model Predictions at Adjacent Timesteps.** Although the difference is relatively large in the low-noise stage, it primarily arises from high-frequency components, which have limited influence on the perceptual quality of the generated results. Different colors indicate different prompts.

## 3.1 PRELIMINARIES

**Rectified Flow** (Liu et al., 2022) models a linear path between the data distribution $\pi_0$ and Gaussian noise $\pi_1$ via an ODE: $d\boldsymbol{Z}_t = v(\boldsymbol{Z}_t, t)dt$, $t \in [0, 1]$, where $v$ is parameterized by a neural network $\boldsymbol{\epsilon_\theta}$. Given samples $\boldsymbol{X}_0 \sim \pi_0$, $\boldsymbol{X}_1 \sim \pi_1$, the forward trajectory is defined by $\boldsymbol{X}_t = (1-t)\boldsymbol{X}_0 + t\boldsymbol{X}_1$, yielding the differential form $d\boldsymbol{X}_t = (\boldsymbol{X}_1 - \boldsymbol{X}_0)dt$. The training objective minimizes the regression loss between the ground truth velocity and the network prediction:

$$\min_\theta \int_0^1 \mathbb{E}\left[\|(\boldsymbol{X}_1 - \boldsymbol{X}_0) - \boldsymbol{\epsilon_\theta}(\boldsymbol{X}_t, t)\|^2\right] dt. \tag{1}$$

At inference, a Gaussian noise $\boldsymbol{Z}_N \sim \mathcal{N}(0, \boldsymbol{I})$ is iteratively updated using the ODE Solver (Lu et al., 2022; Wang et al., 2024c) represented by the Euler Method: $\boldsymbol{Z}_{i-1} = \boldsymbol{Z}_i + (t_{i-1} - t_i)\,\boldsymbol{\epsilon_\theta}(\boldsymbol{Z}_i, t_i)$. Compared to DDPM (Ho et al., 2020), RF achieves high-quality generation with significantly fewer steps due to its linear sampling path. This efficiency makes it well-suited for T2V generation tasks (Zheng et al., 2024; Wang et al., 2025; Yang et al., 2024; Kong et al., 2024).

**Feature Caching.** DiT-based video generation remains computationally intensive due to the complexity of modeling spatiotemporal dependencies and the need for iterative denoising over numerous steps. To address this, feature caching is a widely adopted technique to accelerate video generation, where most works focus on step-wise caching. Considering the noisy latent $\boldsymbol{Z}_i$ at the $i$th denoising step, a full inference of network $\boldsymbol{\epsilon_\theta}$ is performed, i.e., $\boldsymbol{F}_i = \boldsymbol{\epsilon_\theta}(\boldsymbol{Z}_i, t_i)$, and the output $\boldsymbol{F}_i$ is cached. In the subsequent $n$ timesteps $\{t_{i-1}, t_{i-2}, \cdots, t_{i-n}\}$, instead of performing a full inference as $\boldsymbol{F}_{i-k} = \boldsymbol{\epsilon_\theta}(\boldsymbol{Z}_{i-k}, t_{i-k})$ where $k \in \{1, \cdots, n\}$, the cached $\boldsymbol{F}_i$ is reused for updating the noisy latent, i.e., $\boldsymbol{F}_{i-k} = \boldsymbol{F}_i$. Although this vanilla feature caching mechanism achieves significant acceleration, the interval $n$ is fixed. On the other hand, different denoising steps have varying degrees of influence on the final output. Accurately identifying the redundant features within the generation process to achieve adaptive caching remains a challenging problem.

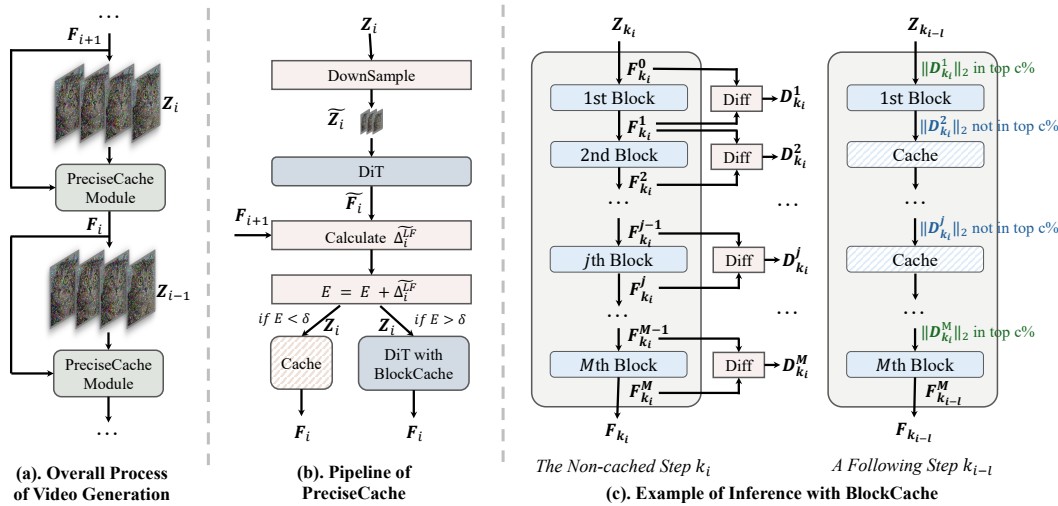

Figure 5: **Pipeline of PreciseCache.**

## 3.2 LOW-FREQUENCY DIFFERENCE

Adaptive caching requires the mechanism to dynamically decide whether to perform a full network inference at each denoising step $t_i$ (i.e., $\mathbf{F}_i = \boldsymbol{\epsilon}_{\boldsymbol{\theta}}(\mathbf{Z}_i, t_i)$), or to reuse the cached predictions of the model from the previous step. Intuitively, this depends on its impact on the final generated videos: if reusing the previous cached prediction at $t_i$ significantly influences the content and quality of the generated video, a full inference of the network $\boldsymbol{\epsilon}_{\boldsymbol{\theta}}$ needs to be conducted; otherwise, the computation of this step can be skipped through reusing the cached prediction.

Based on this intuition, we begin by analyzing the impact of reusing the cached feature at each step on the final generated video. Considering the video generation process consisting of $N$ steps, we respectively skip the computation at each step $t_i$ (where $i \in \{N-1, N-2, \cdots, 0\}$) by reusing the model's prediction from the previous step $t_{i+1}$, and then generate the final videos. We measure the Mean Squared Error (MSE) between videos generated with caching and the ground truth, which are generated without caching. Generally, our results (Figure 3a) indicate that reusing the cached feature at early high-noise steps significantly affects the generated results, whereas at later low-noise steps, its impact is negligible.

The above analysis precisely quantifies the influence of applying caching at each single denoising step. Intuitively, if this influence is estimated immediately at $t_i$ during the denoising process, it can then be leveraged to decide whether the computation at $t_i$ can be skipped through feature caching. However, it is a non-trivial task because the influence of each step in Figure 3a cannot be obtained before the corresponding video is generated. As an alternative, most prior works like (Liu et al., 2025a) directly leverage the difference between the current network prediction $\mathbf{F}_i$ and the cached prediction as the metric to indicate caching (Figure 4a), which does not align with the above observation and would lead to sub-optimal caching strategies. In this work, we propose to further decompose the model prediction $\mathbf{F}_i$ into low-frequency and high-frequency components ($\mathbf{F}_i^{LF}$ and $\mathbf{F}_i^{LF}$) through the Fast Fourier Transform (FFT), and investigate their separate effects during denoising, i.e.,

$$\mathbf{F}_i^{LF} = \mathcal{FFT}(\boldsymbol{\epsilon}_{\boldsymbol{\theta}}(\mathbf{Z}_i, t_i))_{low}; \quad \mathbf{F}_i^{HF} = \mathcal{FFT}(\boldsymbol{\epsilon}_{\boldsymbol{\theta}}(\mathbf{Z}_i, t_i))_{high}. \tag{2}$$

We observe that caching $\mathbf{F}_i^{LF}$ predominantly affects the generated results (Figure 3b), whereas $\mathbf{F}_i^{HF}$ has a negligible influence (Figure 3c). Based on this insight, we further calculate their difference between adjacent denoising steps, i.e.,

$$\Delta_i^{LF} = \left\| \mathbf{F}_i^{LF} - \mathbf{F}_{i+1}^{LF} \right\|_2; \quad \Delta_i^{HF} = \left\| \mathbf{F}_i^{HF} - \mathbf{F}_{i+1}^{HF} \right\|_2, i \in \{N-1, \cdots, 0\} \tag{3}$$

We find that the **Low-Frequency Difference (LFD)** $\Delta_i^{LF}$ closely aligns with the observation in Figure 3a. This implies that at high-noise timesteps, the network generates critical structural and content information for the video, while at low-noise timesteps, it primarily produces high-frequency details that are perceptually insignificant and thus can be safely cached to accelerate generation.

## 3.3 LFCACHE

Directly applying Low-Frequency Difference to indicate caching cannot accelerate the video generation process because obtaining $\Delta_i^{LF}$ requires performing a full forward pass at the timestep $t_i$ to calculate $\boldsymbol{F}_i$. To address this, we propose **LFCache** framework, where a downsampled latent is first fed into the model for "trail" at each denoising step. Specifically, given the latent $\boldsymbol{Z}_i \in \mathbb{R}^{T \times H \times W \times C}$ at the timestep $t_i$, (where $T$, $H$, $W$ and $C$ represent the temporal, height, width, and channel of the latent). We first downsample the latent on its temporal, height and width dimensions, i.e.,

$$\widetilde{\boldsymbol{Z}}_i = \text{Downsample}\,(\boldsymbol{Z}_i), \qquad (4)$$

where $\widetilde{\boldsymbol{Z}}_i \in \mathbb{R}^{(T/r) \times (H/s) \times (W/s) \times C}$, $r$ denotes the downsample factor at the temporal dimen-

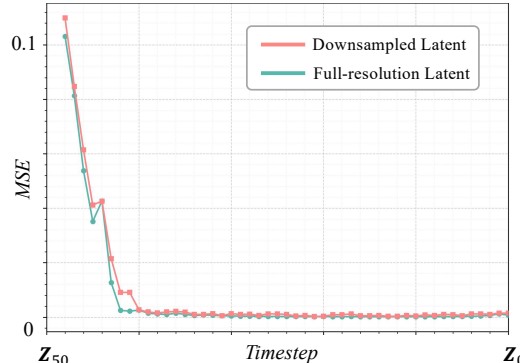

Figure 6: **The Relationship between Latent Resolution and LFD.** We observe that downsampling has little effect on the computation of LFD. The experiment is conducted on Wan2.1-14B (Wang et al., 2025).

sion and $s$ denotes the downsample factor at the spatial dimension. Then, we feed the downsampled latent $\widetilde{\boldsymbol{Z}}_i$ into the network $\epsilon_{\boldsymbol{\theta}}$ to obtain an estimated output $\widetilde{\boldsymbol{F}}_i$, i.e., $\widetilde{\boldsymbol{F}}_i = \epsilon_{\boldsymbol{\theta}}(\widetilde{\boldsymbol{Z}}_i, t_i)$. Due to the reduced size of the downsampled latent, this process is highly efficient, taking a negligible computational overhead within the overall video generation process. Similarly, we downsample the cached prediction $\boldsymbol{F}_{i+1}$, obtaining $\widetilde{\boldsymbol{F}}_{i+1}$ and calculate the low-frequency difference, i.e., $\widetilde{\Delta}_i^{LF} = \left\| \widetilde{\boldsymbol{F}}_i^{LF} - \widetilde{\boldsymbol{F}}_{i+1}^{LF} \right\|_2$. The analysis in Figure 6 indicates that the $\widetilde{\Delta}_i^{LF}$ is highly consistent with $\Delta_i^{LF}$, which can be used as an effective caching indicator during the process of generation.

Following prior works (Liu et al., 2025a), at each denoising step, we use the accumulated differences as the final indicator to indicate caching. Specifically, after doing a full inference and obtaining the output of the model $\boldsymbol{F}_a$ at timestep $t_a$, we accumulate the low-frequency difference at subsequent timesteps, i.e., $\sum_{i=a}^{b} \widetilde{\Delta}_i^{LF}$. If $\sum_{i=a}^{b} \widetilde{\Delta}_i^{LF}$ is greater than a pre-defines threshold $\delta$ at $t_b$, the computation of this timestep cannot be skipped and $\boldsymbol{F}_b$ should be calculated through the inference of the network. Otherwise, we reuse the $\boldsymbol{F}_a$ to update the latent at $t_b$. This procedure is shown in Algorithm 1.

## 3.4 BLOCKCACHE

LFCache effectively identifies which timesteps in the denoising process can be directly skipped by reusing the cached output of the entire DiT model (i.e., $\epsilon_{\boldsymbol{\theta}}$). On the other hand, even at those non-skipped timesteps, redundancy still exists within the computations of individual transformer blocks. To address this and achieve further acceleration, we propose **BlockCache**, which eliminates the redundancy at the block level in those non-skipped steps identified by LFCache. Specifically, considering the non-skipped timestep $t_{k_i} \in \{N, N - k_1, \cdots, N - k_n\}$ identified in section 3.3, the inference of the DiT model with $M$ transformer blocks (i.e., $\boldsymbol{F}_{k_i} = \epsilon_{\boldsymbol{\theta}}(\boldsymbol{Z}_{k_i}, t_{k_i})$) can be decomposed as

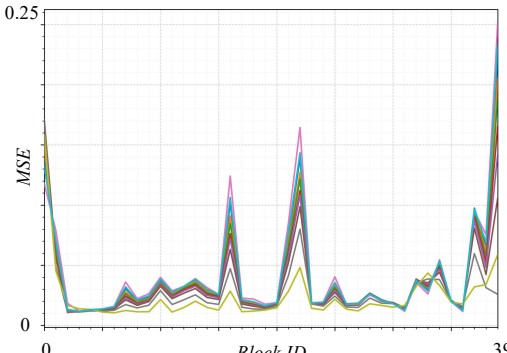

Figure 7: **The Importance of Each Transformer Block within the Video Diffusion Transformer.** Different colors represent different denoising steps. The experiment is conducted on Wan2.1-14B (Wang et al., 2025).

$$\boldsymbol{F}_{k_i}^0 = \boldsymbol{Z}_{k_i}; \quad \boldsymbol{F}_{k_i}^j = \mathcal{B}^j(\boldsymbol{F}_{k_i}^{j-1}, t_{k_i}); \quad \boldsymbol{F}_{k_i} = \boldsymbol{F}_{k_i}^M, \qquad (5)$$

where $j \in \{1, \ldots, M\}$ and $\mathcal{B}^j$ indicates the $j$th block in $\epsilon_{\boldsymbol{\theta}}$. We analyze the redundancy of each block $\mathcal{B}^j$ by calculating the difference between its input and output. The results in Figure 7 illustrate that only a subset of blocks (which we refer to as pivotal blocks) make notable modifications of the input, while remaining blocks have minimal impact (which we refer to as non-pivotal blocks).

---

**Algorithm 1** Video Generation with PreciseCache.

---

1: Initialize $\boldsymbol{\epsilon_\theta}$, $\boldsymbol{Z}_N \sim \mathcal{N}(\boldsymbol{0}, \boldsymbol{I})$
2: Initialize $E \leftarrow 0$                                                   // Accumulated error
3: $\boldsymbol{F}_N \leftarrow \boldsymbol{\epsilon_\theta}(\boldsymbol{Z}_N, t_N)$               // Always do the full inference at the first timestep $t_N$
4: $\widetilde{\boldsymbol{F}}_N^{LF} \leftarrow \text{Downsample}(\mathcal{FFT}(\boldsymbol{F}_N)_{\text{low}})$
5: $\boldsymbol{Z}_{N-1} \leftarrow \text{UpdateLatent}\{\boldsymbol{Z}_N, \boldsymbol{F}_N\}$
6: **for** $i = N-1, N-2, \ldots, 1$ **do**
7:     $\widetilde{\boldsymbol{Z}}_i \leftarrow \text{Downsample}(\boldsymbol{Z}_i)$
8:     $\widetilde{\boldsymbol{F}}_i \leftarrow \boldsymbol{\epsilon_\theta}(\widetilde{\boldsymbol{Z}}_i, t_i)$             // Obtain the network output of downsampled input
9:     $\widetilde{\boldsymbol{F}}_i^{LF} \leftarrow \mathcal{FFT}(\widetilde{\boldsymbol{F}}_i)_{\text{low}}$            // Calculate the low-frequency component at $t_i$
10:    $\widetilde{\boldsymbol{F}}_{i+1}^{LF} \leftarrow \mathcal{FFT}(\text{Downsample}(\boldsymbol{F}_{i+1}))_{\text{low}}$   // Calculate the low-frequency component at $t_{i+1}$
11:    $\widetilde{\Delta}_i^{LF} \leftarrow \left\| \widetilde{\boldsymbol{F}}_i^{LF} - \widetilde{\boldsymbol{F}}_{i+1}^{LF} \right\|_2$       // Calculate the Low-Frequency Difference (LFD)
12:    $E \leftarrow E + \widetilde{\Delta}_i^{LF}$                 // Update the accumulate error
13:    **if** $E < \delta$ **then**
14:       $\boldsymbol{F}_i \leftarrow \boldsymbol{F}_{i+1}$               // Directly reuse the cached output
15:    **else**
16:       $\boldsymbol{F}_i \leftarrow \boldsymbol{\epsilon_\theta}(\boldsymbol{Z}_i, t_i)$          // Inference with **_BlockCache_**
17:       $E \leftarrow 0$                     // Reset error
18:    **end if**
19:    $\boldsymbol{Z}_{i-1} \leftarrow \text{UpdateLatent}(\boldsymbol{Z}_i, \boldsymbol{F}_i)$         // Update latent
20: **end for**
21: **Output:** $\boldsymbol{Z}_0$

---

Based on this observation, our BlockCache aims to eliminate the redundant computation of non-pivotal blocks. Specifically, considering the non-skipped step $t_{k_i}$, a full inference of the network is conducted, and the difference $\boldsymbol{D}_{k_i}^j$ between the input and output of each block is cached, i.e., $\boldsymbol{D}_{k_i}^j = \boldsymbol{F}_{k_i}^j - \boldsymbol{F}_{k_i}^{j-1}$. Then, we select the blocks with top $c\%$ largest difference, which are identified as the pivotal blocks: $\mathcal{I}_{k_i} = \left\{ j \mid \left\| \boldsymbol{D}_{k_i}^j \right\|_2 \text{ is in the top } c\% \text{ of all values} \right\}$. Other blocks are non-pivotal blocks, which are highly redundant and thus can be skipped. In the following $L$ non-skipped denoising steps $t_{k_{i-l}}$ ($l \in \{1, \cdots, L\}$), we use the cached difference $\boldsymbol{D}_{k_i}^j$ to estimate the output of non-pivotal blocks. The inference procedure with BlockCache can be represented as

$$\boldsymbol{F}_{k_{i-l}}^0 = \boldsymbol{Z}_{k_{i-l}}, \quad \boldsymbol{F}_{k_{i-l}}^j = \begin{cases} \mathcal{B}^j\left(\boldsymbol{F}_{k_{i-l}}^{j-1}, t_{k_{i-l}}\right), & j \in \mathcal{I}_i \\ \boldsymbol{F}_{k_{i-l}}^{j-1} + \boldsymbol{D}_{k_i}^j, & j \notin \mathcal{I}_i \end{cases}, \quad \boldsymbol{F}_{k_{i-l}} = \boldsymbol{F}_{k_{i-l}}^N. \tag{6}$$

With the skipping of the non-pivotal blocks, BlockCache minimizes redundancy during generation without compromising the quality of the results. For implementation, our BlockCache is easier to integrate into diverse model architectures for acceleration and only requires minimal hyper-parameter tuning compared to previous block-level caching methods such as (Kahatapitiya et al., 2024).

## 4 EXPERIMENTS

### 4.1 SETUP

**Baselines.** To validate the efficacy of PreciseCache, we implement our method on various state-of-the-art base models for video generation, including Open-Sora 1.2 (Zheng et al., 2024), Hunyuan-Video (Kong et al., 2024), CogVideoX (Yang et al., 2024), and Wan2.1 (Wang et al., 2025). We compare our methods with previous SOTA cached-based acceleration methods for video generation models, including PAB (Zhao et al., 2024b), TeaCache (Liu et al., 2025a), and FasterCache (Lv et al., 2024). For these methods, we utilize their official implementations available on GitHub. For base models not directly supported by their official code, we implement the method ourselves.

**Evaluation Metrics and Datasets.** We evaluate inference efficiency and generated video quality of PreciseCache. To measure the inference efficiency, we report Multiply-Accumulate Operations (MACs) and inference latency. For assessing visual quality, we generate videos using the prompt

from VBench (Huang et al., 2024) and evaluate performance using VBench's comprehensive metrics. We also report some widely adopted perceptual and fidelity metrics, including LPIPS, PSNR, and SSIM, which measure the similarity between videos generated with cache-based acceleration methods and those directly generated by base models without caching.

**Implementation Details.** Determining an appropriate threshold $\delta$ for LFCache is a non-trivial task, as the optimal value tends to vary across different base models and prompts. To address this challenge, we convert determining a specific threshold value into determining a relative factor $\alpha$. Specifically, in our implementation, caching is disabled for the first 5 timesteps during which we record the maximum low-frequency difference observed, i.e., $\widetilde{\Delta}_{max}^{LF}$. We then set the threshold $\delta$ as $\widetilde{\Delta}_{max}^{LF} \times \alpha$. This strategy substantially reduces the difficulty of manually tuning the threshold parameter. For LFCache, we provide two basic configurations, i.e., PreciseCache-Base and PreciseCache-Turbo, where $\alpha$ is set to 0.5 and 0.7 for all the models. Based on PreciseCache-Turbo, we further provide a faster configuration, i.e., PreciseCache-Flash, where the BlockCache is enabled with the cache rate set to 40% and $L$ set to 3. The downsample rate in LFCache is set to [2, 4, 4] in the temporal, height, and width dimensions, respectively. To separate frequency components using FFT, we define a low-frequency region as a centered circular mask with radius equal to $\frac{1}{5}$ of the minimum spatial dimension, i.e., radius $= \frac{1}{5}\min(H, W)$. All experiments are executed on NVIDIA A800 80GB GPUs utilizing PyTorch, with FlashAttention (Dao et al., 2022) enabled by default to optimize computational efficiency.

Table 1: **Quantitative Comparison** of efficiency and visual quality on 4 A800 GPUs.

| Method | Efficiency | | | Visual Quality | | | |
|---|---|---|---|---|---|---|---|
| | MACs (P) ↓ | Speedup ↑ | Latency (s) ↓ | VBench ↑ | LPIPS ↓ | SSIM ↑ | PSNR ↑ |
| **Open-Sora 1.2** (480P, 192 frames) | | | | | | | |
| Open-Sora 1.2 ($T=30$) | 6.30 | 1× | 47.23 | 78.79% | - | - | - |
| PAB | 5.33 | 1.26× | 38.40 | 78.15% | 0.1041 | 0.8821 | 26.43 |
| TeaCache | 3.29 | 1.95× | 24.73 | 78.23% | 0.0974 | 0.8897 | 26.84 |
| FasterCache | 4.13 | 1.67× | 29.15 | 78.46% | 0.0835 | 0.8932 | 27.03 |
| Ours-base | **3.73** | **1.72×** | **27.95** | **78.71%** | **0.0617** | **0.9081** | **28.78** |
| Ours-turbo | **3.10** | **2.07×** | **23.27** | **78.49%** | **0.0786** | **0.8971** | **27.11** |
| Ours-flash | **2.45** | **2.60×** | **18.38** | **78.19%** | **0.0979** | **0.8903** | **26.78** |
| **HunyuanVideo** (480P, 65 frames) | | | | | | | |
| HunyuanVideo ($T=50$) | 14.92 | 1× | 73.64 | 80.66% | - | - | - |
| PAB | 10.73 | 1.35× | 54.54 | 79.37% | 0.1143 | 0.8732 | 27.01 |
| TeaCache | 8.93 | 1.64× | 44.90 | 80.51% | 0.0911 | 0.8952 | 28.15 |
| FasterCache | 10.29 | 1.43× | 51.50 | 80.59% | 0.0893 | 0.9017 | 28.96 |
| Ours-base | **9.15** | **1.61×** | **45.74** | **80.65%** | **0.0654** | **0.9102** | **29.15** |
| Ours-turbo | **7.49** | **1.95×** | **37.76** | **80.49%** | **0.0884** | **0.9043** | **29.06** |
| Ours-flash | **6.04** | **2.44×** | **30.18** | **80.02%** | **0.0902** | **0.8977** | **28.64** |
| **CogVideoX** (480P, 48 frames) | | | | | | | |
| CogVideoX ($T=50$) | 6.03 | 1× | 21.13 | 80.18% | - | - | - |
| PAB | 4.45 | 1.32× | 16.01 | 79.76% | 0.0860 | 0.8978 | 28.04 |
| TeaCache | 3.33 | 1.79× | 11.80 | 79.79% | 0.0802 | 0.9013 | 28.76 |
| FasterCache | 3.71 | 1.60× | 13.21 | 79.83% | 0.0766 | 0.9066 | 28.93 |
| Ours-base | **3.59** | **1.65×** | **12.81** | **80.14%** | **0.0619** | **0.9110** | **29.23** |
| Ours-turbo | **2.96** | **2.02×** | **10.46** | **79.91%** | **0.0742** | **0.9021** | **28.97** |
| Ours-flash | **2.31** | **2.58×** | **8.19** | **79.80%** | **0.0849** | **0.9001** | **28.79** |
| **Wan2.1-14B** (720P, 81 frames) | | | | | | | |
| Wan2.1-14B ($T=50$) | 329.2 | 1× | 907.3 | 83.62% | - | - | - |
| PAB | 233.5 | 1.38× | 657.5 | 82.91% | 0.1853 | 0.8607 | 26.18 |
| TeaCache | 166.3 | 1.94× | 467.7 | 83.24% | 0.1012 | 0.8719 | 27.22 |
| FasterCache | 183.9 | 1.73× | 524.5 | 83.47% | 0.0741 | 0.9078 | 28.45 |
| Ours-base | **204.5** | **1.59×** | **570.6** | **83.56%** | **0.0451** | **0.9189** | **29.12** |
| Ours-turbo | **151.0** | **2.15×** | **422.1** | **83.52%** | **0.0633** | **0.9127** | **28.98** |
| Ours-flash | **122.4** | **2.63×** | **344.9** | **83.43%** | **0.0812** | **0.9035** | **28.76** |

## 4.2 MAIN RESULTS

**Quantitative Evaluation.** Table 1 reports a detailed quantitative assessment comparing our approach with state-of-the-art acceleration methods: PAB (Zhao et al., 2024b), TeaCache (Liu et al., 2025a), and FasterCache (Lv et al., 2024), focusing on both computational efficiency and visual fidelity. Our PreciseCache consistently illustrates notable speedup while strictly maintaining the vi-

*A duck swims in a pond and a model ship floats nearby*

*A woman knits a sweater and a cat plays with the yarn*

*Furry dog fetching a rubber ball*

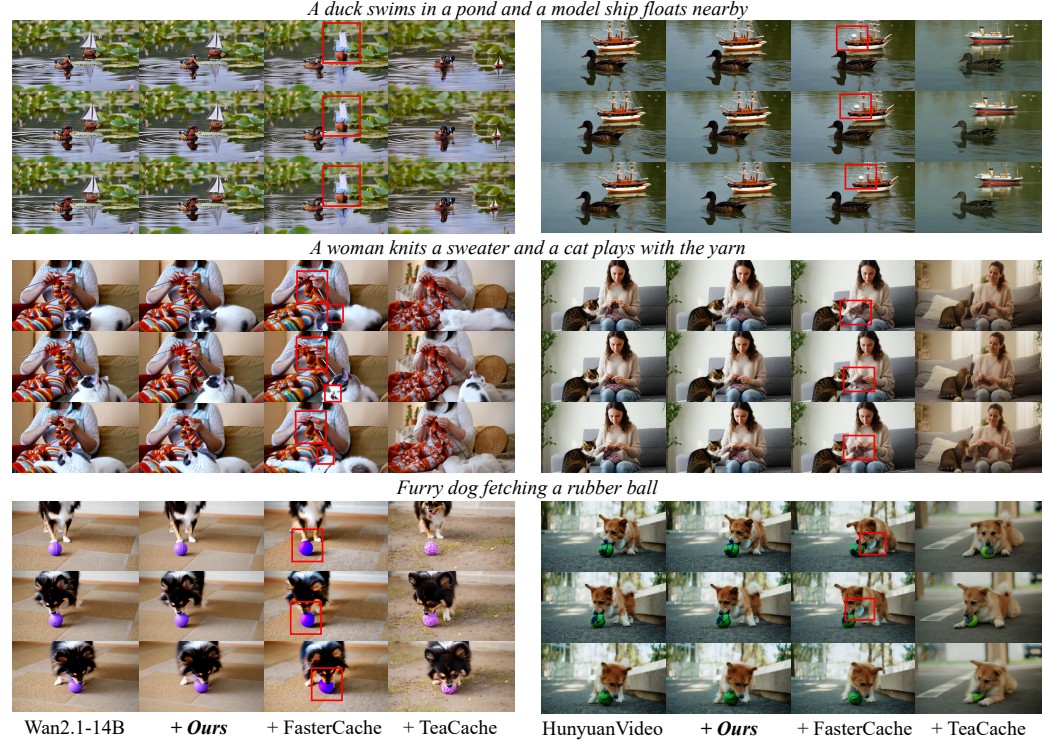

| Wan2.1-14B | **+ *Ours*** | + FasterCache | + TeaCache | HunyuanVideo | **+ *Ours*** | + FasterCache | + TeaCache |

Figure 8: **Qualitative Comparison**. Zoom in for the best views.

sual quality of the base model, demonstrating robustness across diverse base architectures, sampling strategies, video resolutions, and durations.

**Qualitative Comparison.** Figure 8 illustrates qualitative results comparing videos generated using PreciseCache-flash and several baseline methods. Visual comparisons demonstrate that our method achieves significant acceleration without altering the generated video content or compromising quality. In contrast, existing baselines often produce different content and suboptimal quality videos Additional qualitative examples are provided in Figure 9 for further reference.

## 4.3 ABLATION STUDIES

To comprehensively evaluate the effectiveness of PreciseCache, we conduct ablation studies to investigate the performance under different number of GPU, the downsampling size in LFCache, and the feature reusing strategy. Without loss of generality, experiments are conducted on Wan2.1-14B (Wang et al., 2025) and HunyuanVideo (Kong et al., 2024).

Table 2: **Latency on Different Number of GPUs with DSP** (Zhao et al., 2024a). Without loss of generality, we use Wan2.1-14B (Wang et al., 2025) and Hunyuan-Video (Kong et al., 2024) as the base models and generate the 1080P videos, reporting the latency (s) under different numbers of A800 GPUs.

| #GPU | HunyuanVideo | +PreciseCache | Wan-2.1 | +PreciseCache |
|------|--------------|---------------|---------|---------------|
| 1 | 982 (1×) | 470 (2.08×) | 3326 (1×) | 1330 (2.50×) |
| 2 | 566 (1.73×) | 275 (3.57×) | 1732 (1.92×) | 753 (4.41×) |
| 4 | 329 (2.98×) | 161 (6.10×) | 907 (3.67×) | 416 (8.00×) |
| 8 | 175 (5.61×) | 88 (11.16×) | 459 (7.25×) | 229 (14.52×) |

Table 3: **Influence of Downsample Size.** Without loss of generality, experiments are conducted on Wan2.1-14B (Wang et al., 2025) with 4 A800 GPUs.

| Factor ($T \times H \times W$) | Latency ↓ | VBench ↑ | LPIPS ↓ |
|---|---|---|---|
| Baseline | 907 (1×) | 83.62% | - |
| $1 \times 2 \times 2$ | 918 (0.98×) | 83.57% | 0.0797 |
| $1 \times 4 \times 4$ | 525 (1.73×) | 83.49% | 0.0801 |
| $1 \times 8 \times 8$ | 401 (2.26×) | 83.18% | 0.1946 |
| $2 \times 4 \times 4$ | **416 (2.18×)** | **83.52%** | **0.0793** |
| $4 \times 4 \times 4$ | 403 (2.25×) | 83.02% | 0.1875 |

**Performances on Different Number of GPUs.** Following previous works (Lv et al., 2024), we adopt the Dynamic Sequence Parallelism (DSP) to facilitate multi-GPU inference. Table 2 illustrates

the inference latency of PreciseCache-turbo under different numbers of A800 GPUs, where our methods consistently achieves significantly lower inference latency than base models. Notably, PreciseCache-turbo can achieve even further acceleration ratio on Wan2.1-14B (Wang et al., 2025) under fewer number of GPU, e.g., it can achieve about $2.5\times$ acceleration using 1 GPU. These results highlight the effectiveness of our PreciseCache in various number of GPUs.

**Size of Downsampling.** As illustrated in section 3.3, a downsampled latent is fed into the model to obtain the estimated output at each denoising step. We conduct experiments to illustrate the impact of downsampling size (Table 3). Experiments show that a small downsampling ratio results in a large latent size, which significantly increases the inference time. Conversely, over-downsampling can yield predictions that fail to adequately estimate the output at the current timestep, leading to suboptimal caching strategies and degraded video generation quality. Empirically, we find that a sampling rate of $2 \times 4 \times 4$ along the temporal ($T$) and spatial ($H$, $W$) dimensions can achieve a satisfying trade-off between acceleration and generation quality.

**Feature Reusing Strategy.** For the LFCache, we directly store the model's final prediction $\boldsymbol{F}_i$ (i.e., the results after classifier-free guidance) at each non-skipped timestep and reuse this cached prediction in the subsequent skipped steps. On the other hand, we notice that some prior works adopt different feature reusing strategies, such as caching the residual (Liu et al., 2025a) (i.e., $\boldsymbol{R}_i = \boldsymbol{F}_i - \boldsymbol{Z}_i$) at the non-skipped steps $t_i$. At the skipped steps, the prediction is estimated according to this cached residual and the input noisy latent. Some works such as TaylorSeer

Table 4: **Feature Reusing Strategy for Step-wise Caching.** Without loss of generality, we conduct experiments on Wan2.1-14B (Wang et al., 2025), generating videos with 1080P resolution.

| Strategy | VBench ↑ | LPIPS ↓ |
|---|---|---|
| Reuse prediction ($\boldsymbol{F}$) | 83.52% | 0.0793 |
| Reuse residuals ($\boldsymbol{R}$) | 83.50% | 0.0791 |
| TaylorSeer | 83.54% | 0.0801 |

(Liu et al., 2025b) also design more sophisticated reuse strategies with better performance. We conducted experiments to compare these strategies and found that their performances are comparable (Table 4) under our PreciseCache. As a result, we adopt the vanilla approach for simplicity. This observation further implies that designing methods to identify *where* and *when* to cache could be more important than exploring *how* to cache for training-free video generation acceleration.

## 5 CONCLUSION

In this work, we propose PreciseCache, an effective training-free method for accelerating the video generation process, containing LFCache for step-wise caching and BlockCache for block-wise caching. First, we introduce the low-frequency difference, which can precisely reflect the redundancy of each denoising step. Then, we propose LFCache which indicates step-wise caching through the low-frequency difference between the downsampled output at the current step and that of the cached step. Furthermore, we propose the BlockCache to reduce the redundancy at the non-skipped timesteps by caching the blocks which has minimal impact on the input feature. Extensive experiments illustrate the effectiveness of our method with different base models under various numbers of GPUs, highlighting its potential for real-world applications.

ACKNOWLEDGMENTS

This work is partially supported by the National Natural Science Foundation of China (No. 62306261), HK RGC-Early Career Scheme (No. 24211525), ITSP Platform Project (No. ITS/600/24FP)and the SHIAE Grant (No. 8115074). This study was supported in part by the Centre for Perceptual and Interactive Intelligence, a CUHK-led InnoCentre under the InnoHK initiative of the Innovation and Technology Commission of the Hong Kong Special Administrative Region Government. This work is also partially supported by Hong Kong RGC Strategic Topics Grant (No. STG1/E-403/24-N), and CUHK-CUHK(SZ)-GDST Joint Collaboration Fund (No. YSP26-4760949). This work is partially supported by Alibaba Research Intern Program.

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

## A APPENDIX

### A.1 MORE QUALITATIVE RESULTS

We provide more qualitative results of our PreciseCache in Figure 9, illustrating the effectiveness of our method.

### A.2 LIMITATIONS AND FUTURE WORKS

Although PreciseCache can achieve significant acceleration of video generation without training, its BlockCache component requires caching the features of each transformer block, which leads to increased GPU memory usage. Consequently, running PreciseCache-flash with BlockCache on Wan2.1-14B to generate 1080P videos cannot be completed on a single 80G A800 GPU. This issue can be addressed through multi-GPU inference. We notice that the increase in GPU memory usage is a common problem of cache-based acceleration methods for the need to store features, which remains to be explored by future work. The application of PreciseCache on more powerful CFG methods (Karras et al., 2024; Chen et al., 2025b) or other diffusion-based scenarios such as controllable generation (Cao et al., 2025; Lin et al., 2025b; Chen et al., 2025a; Lin et al., 2025a), unified models (Xie et al., 2024; Xiao et al., 2025b;a), detecting Chen et al. (2023) and segmentation (Fang et al., 2024) are also promising directions for future works.

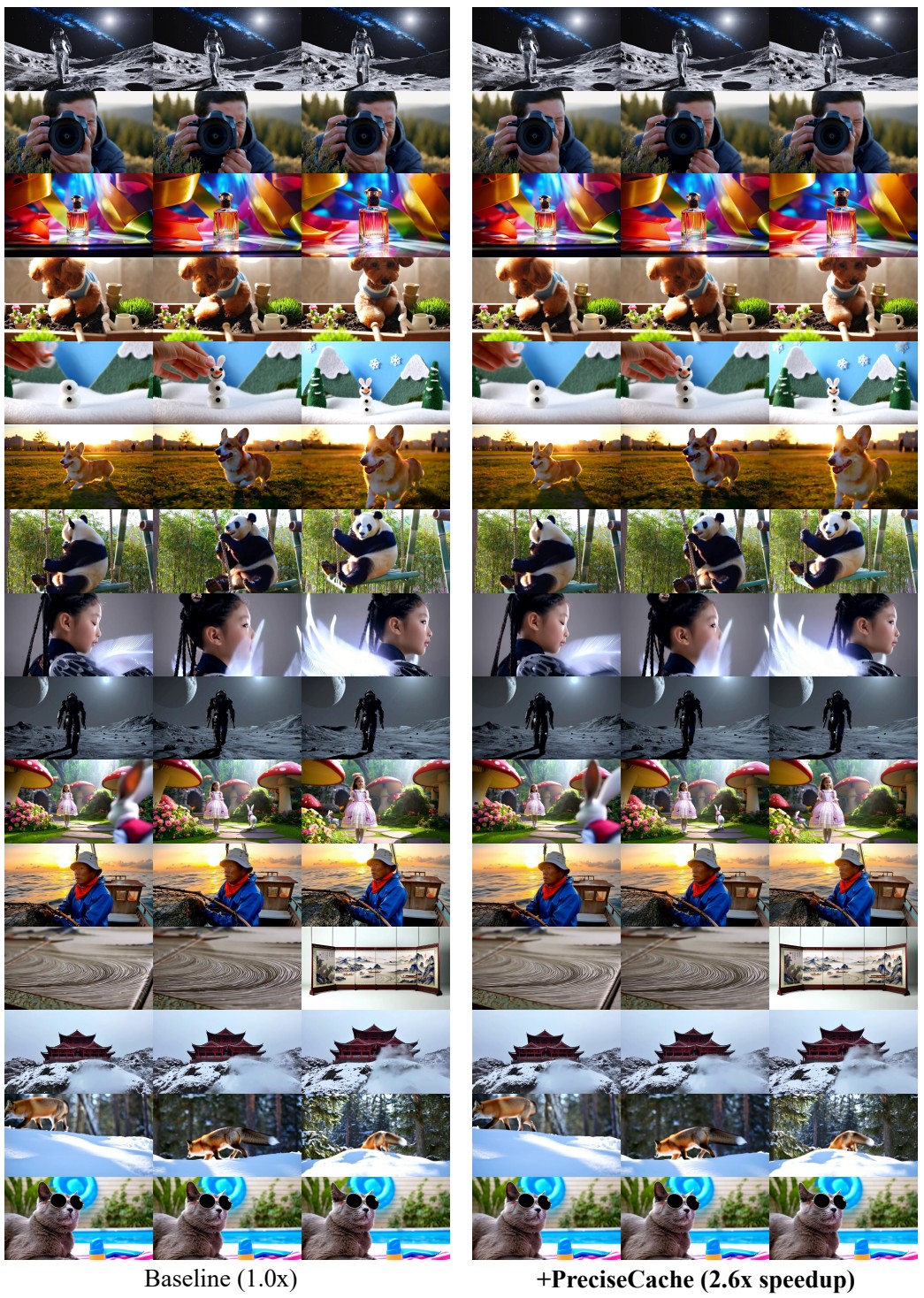

Baseline (1.0x)        +PreciseCache (2.6x speedup)

Figure 9: **More Qualitative Results of PreciseCache on Wan2.1-14B**. Zoom in for the best views.

