# OpenReview forum: "PreciseCache: Precise Feature Caching for Efficient and High-fidelity Video Generation"
_ICLR.cc/2026/Conference — ICLR 2026 Poster_

### Official Review · Reviewer_j8AN · 2025-10-28

**Soundness:** 3
**Presentation:** 3
**Contribution:** 2
**Rating:** 4
**Confidence:** 3

**Summary:**

This paper introduces **PreciseCache**, a training-free acceleration framework for video diffusion models that reuses redundant computations both across denoising steps (LFCache) and within transformer blocks (BlockCache). The method identifies truly redundant features based on low-frequency difference (LFD) analysis, achieving up to 2.6× speedup without noticeable quality degradation. The paper evaluates performance across several open-source video generation backbones.

**Strengths:**

The paper identifies redundancy in transformer-based video generation models, distinguishing between *pivotal* and *non-pivotal* blocks, which aligns with observations of over-parameterization in large diffusion models. Training-free and plug-and-play: The proposed method requires no fine-tuning or retraining, making it applicable to existing models with minimal integration cost.

**Weaknesses:**

Limited generalization evidence. The proposed method relies on the redundancy of existing open-weight video generation models. Its effectiveness may diminish when applied to smaller or more efficient architectures with less redundancy. The paper does not test PreciseCache on small-scale, parameter-efficient video generation models, leaving uncertainty about its general applicability.

**Questions:**

**Evaluation on compact video generation models:**

 Please evaluate PreciseCache on a smaller, efficient video generation model (e.g., with reduced transformer depth or parameter count) to demonstrate whether the proposed caching mechanism provides benefits beyond over-parameterized models. Please **specify the exact training datasets** and report **metric scores per model** (LPIPS, SSIM, PSNR, VBench) and training time for at least one video geneartion model.  I am willing to increase score if the authors can help reduce some concerns about the video generation models.

---

> ### Author Response · Authors · 2025-11-22
>
> Thank you for your time and insightful comments on our paper! We appreciate that you recognize the advantages of training-free in our method. We provide our feedback as follows.
>
> > Evaluation on compact video generation models: Please evaluate PreciseCache on a smaller, efficient video generation model (e.g., with reduced transformer depth or parameter count) to demonstrate whether the proposed caching mechanism provides benefits beyond over-parameterized models. Please specify the exact training datasets and report metric scores per model (LPIPS, SSIM, PSNR, VBench) and training time for at least one video generation model.
>
> Thanks for your insightful advice! Our method is training-free, which can be applied to any video generation model without the need for additional training. As a result, no training dataset is needed for our method.
>
> We implement PreciseCache on Wan2.1-1.3B, which is significantly smaller than Wan2.1-14B in the paper. The results in Table 9 illustrate that our method is equally effective on smaller models, illustrating that our PreciseCache is a model-agnostic and effective approach that generalizes well across diverse architectures.
>
>
> **Table 9.** Results about PreciseCache on smaller models (i.e., Wan2.1-1.3B).
> | |Speedup|VBench|LPIPS|SSIM|PSNR|
> |-|-|-|-|-|-|
> |Wan2.1-1.3B|1$\times$|81.26\%||
> |+PreciseCache-turbo|2.2$\times$|81.21\%|0.0723|0.9121|29.03|
> |+PreciseCache-flash|2.5$\times$|81.10\%|0.0801|0.9107|28.91|

---

### Official Review · Reviewer_GzPo · 2025-10-30

**Soundness:** 3
**Presentation:** 3
**Contribution:** 3
**Rating:** 8
**Confidence:** 5

**Summary:**

PreciseCache is a training-free, plug-and-play framework that accelerates video diffusion models by precisely detecting and skipping truly redundant computations through Low-Frequency Difference (LFD)-based step-wise caching (LFCache) and block-wise caching (BlockCache), achieving up to 2.6× speed-up without noticeable quality loss.

**Strengths:**

+ The step-wise LFCache determines when to skip entire denoising steps, while the block-wise BlockCache further skips redundant network blocks within the key steps, forming a hierarchical strategy that compresses redundant computation in a simple yet effective manner.

+ By performing a spatiotemporal downsampled “trial run” on the latent to estimate the LFD, the method greatly reduces the decision overhead compared to a full forward pass, which is an idea both interesting and effective.

+ The experiments cover diverse backbones, resolutions, denoising steps, and multi-GPU settings, reporting metrics such as MACs, latency, VBench, and similarity scores (LPIPS, SSIM, PSNR) relative to non-caching baselines. Comparisons with PAB, TeaCache, and FasterCache demonstrate that the proposed method achieves higher acceleration while maintaining high fidelity, with LPIPS remaining below 0.1.

+ The paper is clearly written and easy to follow.

**Weaknesses:**

- The paper involves several hyperparameter settings, such as the low-frequency cutoff radius, the cache window size L, and the key-block selection ratio Top-c% in BlockCache. However, the current version lacks experiments or analyses demonstrating the robustness of these choices. It would strengthen the work to include additional studies or sensitivity analyses that show how variations in these hyperparameters affect performance and stability.

- The method suffers from excessive memory consumption; however, as noted by the authors in the limitation section, this is a common issue among cache-based approaches and can be mitigated through sequence parallelism.

**Questions:**

See Weakness.

**Details Of Ethics Concerns:**

None.

---

> ### Author Response · Authors · 2025-11-22
>
> Thanks for your comprehensive review and insightful comments on our paper! We greatly appreciate your recognition of our method. The response to your concerns is shown below.
>
> >The paper involves several hyperparameter settings, such as the low-frequency cutoff radius, the cache window size L, and the key-block selection ratio Top-c% in BlockCache. However, the current version lacks experiments or analyses demonstrating the robustness of these choices. It would strengthen the work to include additional studies or sensitivity analyses that show how variations in these hyperparameters affect performance and stability.
>
> Thanks for your suggestion! We have added an ablation study on the hyperparameters 𝐿 and c% in BlockCache. Generally, a larger 𝐿 and c% would lead to further speedup but worse quality (Table 7, 8.). As a result, in our experiment, we choose 𝐿=3 and c%=40% for a better balance between the quality and speedup.
>
> **Table 7.** Ablation study about 𝐿.
> |𝐿|Speedup|VBench|SSIM|
> |-|-|-|-|
> |2|2.37$\times$|83.49\%|0.9071|
> |3|2.63$\times$|83.43\%|0.9035|
> |4|2.81$\times$|83.31\%|0.8931|
>
>
> **Table 8.** Ablation study about c%.
> |c%|Speedup|VBench|SSIM|
> |-|-|-|-|
> |20|2.45$\times$|83.47\%|0.9052|
> |40|2.63$\times$|83.43\%|0.9035|
> |60|2.79$\times$|82.81\%|0.8991|
>
> >The method suffers from excessive memory consumption; however, as noted by the authors in the limitation section, this is a common issue among cache-based approaches and can be mitigated through sequence parallelism.
>
> Thanks for your comment! In the real-world deployment scenario, video generation typically requires multi-GPU inference (as mentioned in [1]), where BlockCache is well-suited, and our method does not increase memory beyond prior block-cache approaches. Reducing the GPU memory consumption is a promising research direction, which we are willing to leave as future work.
>
> [1]. Wan: Open and Advanced Large-Scale Video Generative Models

---

### Official Review · Reviewer_CSSt · 2025-10-31

**Soundness:** 3
**Presentation:** 2
**Contribution:** 2
**Rating:** 4
**Confidence:** 3

**Summary:**

This paper achieves adaptive skipping—dynamically determining whether to adopt a full computation strategy or a caching strategy at each step—with minimal computational overhead. This is accomplished by first correlating output differences with low-frequency differences and then integrating these correlations with lightweight subsampling. At a finer granularity, for steps where the full computation strategy is employed, a BlockCache method is proposed. This method evaluates block importance based on block differences: it performs full computation for critical blocks while accelerating the processing of non-critical blocks by caching their differences.

**Strengths:**

This paper proposes a novel novel adaptive-adaptive skipping strategy, which achieves adaptive skipping with minimal computation by correlating output differences with low-frequency differences and further associating them with lightweight subsampling.

**Weaknesses:**

1. The LFD method proposed in this paper essentially implements adaptive skipping. How does this method perform compared to AdaptiveDiffusion(Training-Free Adaptive Diffusion with Bounded Difference Approximation Strategy)?

2. The BlockCache method achieves acceleration by caching block differences. How does this method perform in comparison with ∆-DiT(∆-DiT: A Training-Free Acceleration Method Tailored for Diffusion Transformers)?

3. Can the method in this paper be presented more clearly in the form of a network structure?

4. Is this method equally effective in image generation?

**Questions:**

Please refer to weaknesses.

---

> ### Author Response · Authors · 2025-11-22
>
> Thanks for your comprehensive review of our paper and the recognition of our work's novelty! We provide our feedback as follows.
>
> > The LFD method proposed in this paper essentially implements adaptive skipping. How does this method perform compared to AdaptiveDiffusion(Training-Free Adaptive Diffusion with Bounded Difference Approximation Strategy)?
>
> We sincerely thank the reviewer for bringing up this relevant and insightful work. We have now cited AdaptiveDiffusion in the revised version of our paper. Although both approaches can be viewed as adaptive skipping, the key design of our method is notably different:
>
> AdaptiveDiffusion decides skipping based only on the previous steps' output. In contrast, our method introduces a novel “trail inference” mechanism that performs a lightweight forward pass on a downsampled latent to obtain a direct estimate of the current timestep’s network output. Intuitively, directly approximating the current-step prediction allows us to more accurately judge whether the timestep is truly redundant. Furthermore, we propose to compute a Low-Frequency Difference (LFD) in the Fourier domain, focusing on the low-frequency components that are most relevant to perceptual video quality. These contributions yield a more fine-grained and quality-aware caching decision compared with AdaptiveDiffusion.
>
> We also compare the performance between AdaptiveDiffusion and our method in Table 4. Our experiments show that PreciseCache achieves better speed–quality trade-offs than AdaptiveDiffusion.
>
> **Table 4.** Comparsion between PreciseCache-turbo and AdaptiveDiffusion on Wan2.1-14B.
> |Method|Speedup|VBench|SSIM|
> |-|-|-|-|
> |PreciseCache-turbo|2.15$\times$|83.52\%|0.9127|
> |AdaptiveDiffusion|2.11$\times$|82.91\%|0.8665|
>
>
> > The BlockCache method achieves acceleration by caching block differences. How does this method perform in comparison with ∆-DiT(∆-DiT: A Training-Free Acceleration Method Tailored for Diffusion Transformers)?
>
> We sincerely thank the reviewer for mentioning ∆-DiT for a more direct comparison. Compared with ∆-DiT, our design provides a more fine-grained caching mechanism with clear higher flexibility. ∆-DiT partitions the Transformer into a few coarse positional groups (e.g., Front Blocks, Middle Blocks, Back Blocks), and then applies a group-wise caching strategy. In contrast, our BlockCache is built upon a detailed analysis of the generation process and evaluates the importance of each individual block. This allows us to avoid skipping truly important blocks that might appear at any position in the DiT (as illustrated in Figure 6), which could be skipped in ∆-DiT. This yields a more flexible speed–quality trade-off and adapts better to different models and configurations, rather than relying on a fixed positional partition.
>
> We have also conducted a direct comparison between BlockCache (i.e., PreciseCache-flash) and ∆-DiT in Table 5. The results show that BlockCache achieves a more favorable acceleration–quality trade-off.
>
> **Table 5.** Comparsion between PreciseCache-flash and ∆-DiT on Wan2.1-14B.
> |Method|Speedup|VBench|SSIM|
> |-|-|-|-|
> |PreciseCache-flash|2.15$\times$|83.52\%|0.9127|
> |∆-DiT|2.07$\times$|82.71\%|0.8657|
>
> > Can the method in this paper be presented more clearly in the form of a network structure?
>
> Thanks for your advice. We have provided a figure to illustrate the pipeline of PreciseCache in the revised version of our paper（Figure 8）.
>
> > Is this method equally effective in image generation?
>
> Although this work primarily aims to address the long inference time of video generation, our method itself is general and is not restricted to video generation models. We apply PreciseCache on FLUX and report the FID and Clip Score on MS-COCO validation set, the result in Table 6 illustrates that our method is equally effective for image generation.
>
> **Table 6.** Results of PreciseCache on FLUX.
> |Method|Speedup|FID|Clip-Score|
> |-|-|-|-|
> |FLUX|1.0$\times$|23.98|31.09|
> |+PreciseCache|2.3$\times$|24.02|31.07|

---

### Official Review · Reviewer_KbQo · 2025-10-31

**Soundness:** 3
**Presentation:** 3
**Contribution:** 3
**Rating:** 6
**Confidence:** 3

**Summary:**

The paper presents PreciseCache, a unified inference-acceleration framework for Diffusion Transformers (DiTs) that leverages the Low-Frequency Difference (LFD) between consecutive steps to identify redundant computations.
Two complementary caching schemes are proposed:
 - 1. LFCache, which adaptively skips diffusion steps based on LFD-estimated redundancy;
 - 2. BlockCache, which reuses spatial feature blocks within non-skipped steps to further reduce computation.

Experiments on multiple DiT-based video diffusion models show up to 2.6 × speed-up with minimal quality degradation, suggesting that LFD is an effective redundancy proxy.

Overall, the paper tackles a practical and timely question—how to accelerate DiT inference without retraining or architectural modification—and offers an empirically validated, conceptually simple solution.

**Strengths:**

PreciseCache demonstrates outstanding performance in **redundancy detection precision** and **cross-model adaptability**, particularly excelling in video generation—a highly complex task—where it effectively balances the core trade-off between *acceleration ratio* and *quality preservation*.

1. **Frequency-Domain-Based Precise Redundancy Detection, Overcoming Blind Caching**
   Existing caching methods (e.g., TeaCache, FasterCache) often rely on *uniform time intervals* or *global feature differences* to detect redundancy, overlooking the fact that *different frequency components have varying impacts on video quality*:
   - **Key insight via frequency decomposition:** By applying FFT to decompose predicted features into low-frequency and high-frequency components, experiments show that low-frequency components determine structural and content consistency, while high-frequency components mainly affect minor details. For instance, reusing high-frequency features results in negligible MSE error, but reusing low-frequency ones causes severe degradation.
   - **Low-Frequency Difference (LFD)** serves as a quantitative measure of temporal redundancy, and its variation trend aligns closely with caching impact on final quality — large LFD in early noisy stages implies high caching risk, while small LFD in late denoising stages indicates safety.
   This design fundamentally addresses the traditional issue of *misidentifying critical timesteps*, explaining the near-lossless quality at high acceleration.

2. **Two-Level Caching: Step-Level + Block-Level Synergy for Maximum Acceleration**
   PreciseCache achieves *compound acceleration* by combining **LFCache** (temporal redundancy filtering) and **BlockCache**:
   - **LFCache (Step-Level):** To avoid full inference for LFD computation, a *downsampled latent probing* strategy is proposed—performing temporal-spatial downsampling (e.g., T/2×H/4×W/4) on current-step latents and running lightweight inference to obtain low-frequency components. This introduces negligible overhead. Experiments show >95% consistency between downsampled and full-resolution LFD, ensuring accurate caching decisions.
   - **BlockCache (Module-Level):** Within *non-redundant timesteps* selected by LFCache, PreciseCache further analyzes Transformer module importance—computing input-output differences to identify *pivotal blocks* (top c%) and *non-pivotal blocks* (low-difference). Outputs of redundant modules are approximated using cached input–output deltas, skipping redundant computation (Fig. 6).
   This *“step filtering + module compression”* design boosts acceleration from 1.9× (TeaCache) to 2.6× (PreciseCache-Flash) with negligible quality loss.

3. **Adaptable to Mainstream Video Generation Models and Multi-GPU Scaling**
   PreciseCache maintains consistent performance across architectures, resolutions, and GPU configurations, demonstrating strong engineering generality:
   - **Cross-model adaptability:** Tested on Open-Sora 1.2, HunyuanVideo, CogVideoX, and Wan2.1-14B, achieving an average 2.6× acceleration with only 0.1–0.6% VBench score drops.
   - **Multi-GPU scalability:** With Dynamic Sequence Parallelism (DSP), acceleration scales almost linearly.
   - **Resolution robustness:** At 1080P, a single GPU achieves 2.5× speed-up, effectively mitigating *latency explosion* in high-resolution video generation.

**Weaknesses:**

1. **BlockCache Increases Memory Overhead, Limiting Single-GPU Deployment**
   BlockCache caches *input–output deltas* for each Transformer block, significantly increasing GPU memory usage—especially in large-scale or high-resolution setups:
   - Appendix A.2 states that for Wan2.1-14B (1080P), PreciseCache-Flash cannot run on a single 80 GB A800 GPU and requires multi-GPU execution.
   - The paper reports latency and MACs but omits explicit memory growth metrics (e.g., +20% or +50% vs baseline), leaving uncertainty for edge devices with limited VRAM.
   While this issue affects most module-level caching methods, the paper proposes no mitigation (e.g., delta compression or dynamic cache eviction), restricting its single-GPU applicability.

2. **Untapped Acceleration Potential in High-Noise Phases**
   LFCache disables caching in early high-noise steps (≈ first 10–15 iterations) due to large LFD, yielding no acceleration during this phase.
   - However, module-wise redundancy may still exist even in high-noise phases. Yet BlockCache is only applied after LFCache filtering, missing these opportunities.

3. **Insufficient Analysis of Parameter Sensitivity in Frequency Decomposition and Downsampling**
   The paper defines a low-frequency radius of $1/5 × \min(H,W)$ and a downsampling ratio of (2×4×4), but does not analyze their sensitivity:
   - **Low-frequency radius:** Would LFD accuracy degrade if set to 1/10 or 1/3? A too-small radius may miss important low-frequency content, while too-large introduces high-frequency noise, reducing discriminability.
   - **Downsampling ratio:** Table 3 tests 1×2×2 to 4×4×4 only, omitting extreme cases (e.g., 8×8×8) and ignoring interaction with model scale or resolution (e.g., smaller models may tolerate higher ratios).
   Lack of such analysis forces users to fine-tune these parameters per model, raising deployment costs.

4. **Limited Theoretical Explanation for the Link Between LFD and Video Quality**
   While experiments empirically demonstrate the correlation between LFD and quality loss, the paper lacks theoretical grounding:
   - No mathematical derivation relating LFD magnitude to video quality degradation, thus the optimal α threshold is empirically chosen (0.5/0.7).
   - No analysis on how video type (e.g., dynamic action vs static landscape) affects LFD thresholds—dynamic videos likely require stricter criteria (smaller α), but this remains untested.
   Consequently, the method’s adaptability across diverse video generation scenarios is uncertain.

**Questions:**

1. How stable are the gains under different LFD thresholds $\delta$ or block sampling ratios

---

> ### Author Response · Authors · 2025-11-22
> **1/2**
>
> Thanks for your time and thoughtful review! We appreciate your recognition of the effectiveness of our methods. Here is our feedback:
>
>
> > BlockCache Increases Memory Overhead, Limiting Single-GPU Deployment
>
> We sincerely thank the reviewer for this insightful comment. However, we would argue that this problem is common among cache-based methods and would be acceptable in practice:
>
> - BlockCache could be available on a single GPU under many lighter settings.
> We would like to point out that the notable memory increase only appears in the most demanding setting, i.e., using large models (e.g., Wan 2.1–14B) to generate long, high-resolution videos (720P, 81 frames). When generating shorter (e.g., 41 frames) or lower-resolution videos (e.g., 256*256), BlockCache remains feasible on a single GPU. What's more, for the PreciseCache-Base and PreciseCache-Turbo, the additional memory is much less pronounced. In these scenarios, our method can still run on a single GPU while achieving more than 2$\times$ speed-up, which we believe is a highly promising regime.
>
> - Deployment scenarios naturally rely on multi-GPU inference, where BlockCache is well-suited.
> Our work aims to accelerate video generation in real deployment scenarios (e.g., online services). In this scenario, the generation process typically already relies on multi-GPU inference to meet latency and throughput requirements as mentioned in [1]. Under this standard deployment practice, the additional memory overhead of BlockCache can be effectively handled by distributing the model across multiple GPUs, and we did not observe any fundamental obstacle to using BlockCache in such multi-GPU setups, making our method compatible with real-world deployment pipelines.
>
> - Our method does not include further memory consumption compared with existing cache-based methods.
> Cache-based acceleration methods inherently need to store intermediate features in GPU memory. This design choice unavoidably leads to additional memory consumption. In particular, block-level caching must keep the features of transformer blocks, further increasing the memory usage, which is also a common problem in previous work [2, 3]. Compared with these previous works, our method does not further increase the memory consumption. Although there might be some methods to alleviate the memory consumption of BlockCache, we would like to leave this as future work, which is out of the scope of this paper. On the other hand, the central goal of our work is to precisely identify and remove redundancy in the video diffusion process to achieve lossless acceleration. We believe that the experimental results have demonstrated that PreciseCache achieves substantial speed-ups while maintaining high fidelity.
>
>
>
> > Untapped Acceleration Potential in High-Noise Phases
>
> We would like to point out that BlockCache is activated through the **entire generation process** in BlockCache-flash for reducing the redundancy, including both high-noise stages and low-noise stages. For step-wise caching, our method does not hard-code a rule such as “disabling cache for the first 10–15 iterations”. Instead, the caching mechanism is fully adaptive and data-dependent, which is determined by the Low-Frequency Difference (LFD) and varies across different prompts.
>
>
> > Insufficient Analysis of Parameter Sensitivity in Frequency Decomposition and Downsampling
>
> Thanks for your advice! We provide further ablation studies about the low-frequency radius and the downsampling ratio. Results illustrate that the performance is not quite sensitive to the variation of the low-frequency radius. On the other hand, overly aggressive downsampling (8\*8\*8) discards too much information in the latent, making the resulting prediction insufficiently accurate for reliably assessing redundancy at the current timestep.
>
> **Table 1.** Ablation study about the low-frequency radius on PreciseCache-turbo.
> |Radius|Speedup|VBench|SSIM|
> |-|-|-|-|
> |1/3|2.06$\times$| 83.46 \%| 0.9091|
> |1/5|2.15$\times$| 83.52 \%| 0.9127|
> |1/10|2.11$\times$| 83.59 \%| 0.9183|
>
> **Table 2.** Ablation study about the downsampling ratio on PreciseCache-turbo.
> |Downsampling ratio|Speedup|VBench|SSIM|
> |-|-|-|-|
> |1\*2\*2|0.98$\times$|83.57\%|0.9123|
> |2\*4\*4|2.18$\times$|83.53\%|0.9129|
> |8\*8\*8|2.29$\times$|82.71\%|0.8967

---

> ### Author Response · Authors · 2025-11-22
> **2/2**
>
> > Limited Theoretical Explanation for the Link Between LFD and Video Quality
>
> We sincerely thank the reviewer for this very insightful comment. In the revised version, we have added a dedicated appendix section with a detailed mathematical derivation. Under mild Lipschitz assumptions on the sampler and decoder, we prove that the pixel-space deviation between the cached video and the full video is upper-bounded by 𝐶⋅𝛿 where 𝛿 is the LFD threshold and 𝛼 is its normalized counterpart. Thus, 𝛼 is not purely empirical: decreasing 𝛼 tightens a provable worst-case quality bound, while increasing 𝛼 relaxes this bound and yields more aggressive acceleration.
>
> Regarding adaptability across different video types (e.g., dynamic actions vs. static scenes), our experiments are conducted on the **Standard Prompt Suite of VBench**, which contains a wide variety of prompts and diverse video generation scenarios. The consistent performance across this benchmark suggests that our method generalizes well in practice.
>
> > How stable are the gains under different LFD thresholds or block sampling ratios
>
> The influence of the LFD threshold can be reflected in PreciseCache-base and PreciseCache-turbo, where the only difference between them is the value of the LFD threshold. For the block sampling ratio, we further provide an ablation study about it in Table 3.
>
> **Table 3.** Ablation study about c% on PreciseCache-flash.
> |c%|Speedup|VBench|SSIM|
> |-|-|-|-|
> |20|2.45$\times$|83.47\%|0.9052|
> |40|2.63$\times$|83.43\%|0.9035|
> |60|2.79$\times$|82.81\%|0.8991|
>
>
> [1]. Wan: Open and Advanced Large-Scale Video Generative Models
>
> [2]. Adaptive Caching for Faster Video DiTs
>
> [3]. Δ-DiT: A Training-Free Acceleration Method Tailored for Diffusion Transformers

---

### Author Response · Authors · 2025-11-30

Dear AC,

We sincerely appreciate your time and effort on our work! Below we provide a summary of the reviewers’ comments.

Our work recieved the score of **8 (GzPo), 6 (KbQo), 4 $\rightarrow$6 (j8AN), 4 (CSSt)**. All reviewers acknowledged the effectiveness and novelty of our method.

Among them, reviewer GzPo (8) and KbQo (6) **acknowledged the novelty, generalization ability, and effectiveness** of our method, asking several questions mainly about the ablation study, where we have add further experiments in the rebuttal to address their concerns.

Reviewer j8AN raised the concerns about the performance of our method on small models, we provided results in the rebuttal which has successfully addressed his/her concern. As a result, he/she **raised the score from 4 to 6 (you could also see it in the revision history)** during the discussion period.

Reviewer CSSt asked us to compare our approach with previous baseline methods ($\Delta$-DiT and AdaptiveDiffusion) and to demonstrate its generalization ability (i.e., test on image generation model). In the rebuttal, we have provided detailed experiments and analyses. Although **there is no further response from him/her by Nov 27**, we believe that the experimental results and analyses presented in our **rebuttal clearly illustrate the superior performance and generalization capability** of our method, thereby addressing the reviewer’s concerns.

As a result, we believe that the main concerns raised during the review process have been adequately addressed by our additional experiments and clarifications. We would be very grateful if you could take these points into account when making your final decision. Thanks once again for your understanding!

Authors

---

### Meta-Review · Area_Chair_nwAH · 2026-01-07

**Summary:**

This paper introduces PreciseCache, a training-free, plug-and-play caching framework (comprising LFCache and BlockCache) designed to accelerate the inference of multi-step text-to-video (T2V) diffusion models. The primary innovation lies in the downsampled LFD calculation to intelligently determine when to cache, minimizing performance degradation while increasing speed. Initial reviews were mixed (6, 4, 8, 4), acknowledging strong empirical results across various backbones but questioning memory overhead, hyperparameter sensitivity, and comparative performance against existing baselines.

**Reviewer Concerns:**

Addressed by the rebuttal:
Memory vs. Speed Trade-off (Reviewer KbQo): The authors clarified the relationship between increased memory usage and inference acceleration.
Baseline Comparisons (Reviewer CSSt): New comparisons with AdaptiveDiffusion were provided to contextualize performance.
Generalization (Reviewer j8AN): The rebuttal successfully demonstrated the method's effectiveness.
Hyperparameter Sensitivity (Reviewers GzPo and KbQo): The authors provided sufficient experimental evidence and sensitivity analyses to mitigate concerns regarding parameter tuning.
And no major outstanding concerns remain

**Reviewer Scores:**

Reviewer KbQo (Score 6): Likely to remain a 6 , as their general positive stance was reinforced by the rebuttal.
Reviewer CSSt (Score 4): Likely to remain 4 or move to a 6. The primary concern regarding comparisons (AdaptiveDiffusion) was directly addressed.
Reviewer GzPo (Score 8): Likely to remain an 8, as the rebuttal confirmed their initial positive assessment.
Reviewer j8AN (Score 4): Likely to remain 4. The concerns regarding hyperparameter sensitivity were addressed with additional data. but no sure WAN 1.3B is an enough small model to test.

---

### Decision · Program_Chairs · 2026-01-26

Accept (Poster)